# Association between Polyunsaturated Fatty Acid and Reactive Oxygen Species Production of Neutrophils in the General Population

**DOI:** 10.3390/nu12113222

**Published:** 2020-10-22

**Authors:** Nobuaki Suzuki, Kaori Sawada, Ippei Takahashi, Motoko Matsuda, Shinji Fukui, Hidemasa Tokuyasu, Hiroyasu Shimizu, Junichi Yokoyama, Arata Akaike, Shigeyuki Nakaji

**Affiliations:** 1Department of Cardiovascular Surgery, Aomori Prefectural Central Hospital, 2-1-1 Higashi Tsukurimichi, Aomori 030-8553, Japan; 2School of Medicine, Hirosaki University, 5 Zaifu-cho, Hirosaki, Aomori 036-8562, Japan; iwane@hirosaki-u.ac.jp (K.S.); ippei@lyremizoguchi.com (I.T.); matsuda@ouhs.ac.jp (M.M.); fukui@shokei.ac.jp (S.F.); tokuyasu@tau.ac.jp (H.T.); shimizu@27complete.jp (H.S.); yokoyama@nittai.ac.jp (J.Y.); arata_akaike@env.go.jp (A.A.); ayamiwa@hotmail.com (S.N.)

**Keywords:** polyunsaturated fatty acids, arachidonic acid, dihomo gamma linolenic acid, reactive oxygen species, cross-sectional study

## Abstract

Little is known about the relationship between polyunsaturated fatty acids (PUFAs) and reactive oxygen species (ROS) in the general population. Therefore this study aimed to describe the association of PUFAs with ROS according to age and sex in the general population and to determine whether PUFA levels are indicators of ROS. This cross-sectional study included 895 participants recruited from a 2015 community health project. Participants were divided into 6 groups based on sex and age (less than 45 years old (young), aged 45–64 years (middle-aged), and 65 years or older (old)) as follows: male, young (*n* = 136); middle-aged (*n* = 133); old (*n* = 82); female, young (*n* = 159); middle-aged (*n* = 228); and old (*n* = 157). The PUFAs measured were arachidonic acid (AA), dihomo gamma linolenic acid (DGLA), AA/DGLA ratio, eicosapentaenoic acid (EPA), and docosahexaenoic acid (DHA). ROS considered in the analysis were basal ROS and stimulated ROS levels. Multiple linear analyses showed: (1) significant correlations between PUFA levels, especially DGLA and AA/DGLA ratio, and neutrophil function in the young and middle-aged groups; (2) no significant correlations in old age groups for either sex. Because PUFAs have associated with the ROS production, recommendation for controlled PUFA intake from a young age should be considered.

## 1. Introduction

Globally, ischemic heart disease (IHD), stroke, and cancer are the leading causes of morbidity and mortality in the past 15 years [1]. It is known that elevated level of reactive oxygen species (ROS) are associated with IHD, stroke, and cancer, because of interlinked similar risk factors, such as atherosclerosis, hypertension, diabetes mellitus, and hyperlipidemia [2,3,4,5,6,7,8]. ROS production is a neutrophil function and ROS play an important role as signaling messengers under inflammatory conditions and in the immune system [9]. Thus, the influence of ROS production on the body is important. We reported on two functions of neutrophils in previous studies targeting the general population and athletes [10,11]. The neutrophils functions are: (1) basal ROS (BROS) production, considered a reflection of oxidative stress under normal conditions without any stimulation [12] and (2) stimulated ROS (SROS) production, considered a reflection of oxidative stress under stimulation by a foreign substance [13].

Several experimental studies show that omega-3 polyunsaturated fatty acids (PUFAs), including eicosapentaenoic acid (EPA) and docosahexaenoic acid (DHA), play an important role in some diseases [14,15,16,17]. For example, several studies report that omega−3 PUFAs reduces the risk of cardiovascular disease and improve the efficacy and tolerability of cancer chemotherapy drugs [14,15,16]. Most prospective cohort studies do not find significant associations between omega-6 PUFAs, including arachidonic acid (AA) and dihomo gamma linolenic acid (DGLA), and some cardiovascular diseases [18]. However DGLA can be further desaturated to AA by delta-5 desaturase (D5D) and the ratio of serum AA to DGLA predicts D5D activity [19], which is associated with reducing arterial stiffness [20] and promoting anti-cancer activity [21].

Little is known about the relationship between PUFAs, especially DGLA and the AA/DGLA ratio reflecting D5D activity, and neutrophil function in the general population. Therefore, the aim of this study was to evaluate the association between PUFAs, particularly DGLA levels and AA/DGLA ratio, and neutrophil function according to age and sex in the general population.

## 2. Materials and Methods

### 2.1. Study Design and Subjects

Our study was a cross-sectional analysis of data from the 2015 Iwaki Health Promotion Project. The Iwaki region of Hirosaki City is in the Aomori Prefecture, Northern Japan. A total of 1113 adults (431 men and 682 women) participated in the project. Participants ranged from 20 to 91 years of age with a mean age of 54.5 ± 14.2 years. We excluded participants with current or past use of drugs that influence neutrophil function and ROS. Specifically, the exclusion criteria were: (1) history of hypertension, hyperlipidemia, diabetes mellitus, allergies, cerebrovascular disease, cardiovascular disease, post-orthopedic surgery, rheumatoid arthritis, schizophrenia, or depression; (2) currently taking any steroid, anti-inflammatory drug, anti-allergy drug, sex steroid hormone, anti-rheumatic drug, antibiotic, osteoporosis medication, laxative, or proton-pump inhibitor; or (3) missing data. In total, 895 participants (351 males and 544 females) were included in the analyses (Figure 1). Participants were divided into 6 groups based on sex (male/female) and age group, including those less than 45 years old (young), 45–64 years (middle-aged), and 65 years or older (old). The resulting groups were: male, young (*n* = 136); male, middle-age (*n* = 133); male, old (*n* = 82); female, young (*n* = 159); female, middle-age (*n* = 228); and female, old (*n* = 157). We prepared this study according to the Strengthening the Reporting of Observational Studies in Epidemiology Statement (STROBE) checklist [22]. All participants provided written informed consent, and the institutional review board of the Hirosaki University Graduate School of Medicine’s Medical Research Ethics Committee approved the study (Permission Number: 2014-377). This study was conducted in accordance with the Declaration of Helsinki.

### 2.2. Self-Administered Questionnaire

The participants completed self-administered questionnaires before the day of the study when an interview was conducted. The data collected included age, sex, smoking habit, medical history, and drug use. All of the participants were measured for height and weight. Body mass index (BMI) was calculated using a standard formula (weight (kg)/height (m)^2^).

### 2.3. PUFA Levels

To assess PUFA levels, blood samples were collected the morning of the study while the participants were in a fasting state started at 9 p.m. of the previous day. The total white blood cell and neutrophil counts were measured using an automated blood cell counter (LSI Medience Corporation, Tokyo, Japan). The serum EPA, DHA, AA, and DGLA levels were measured by an external laboratory (LSI Medience Corporation, Tokyo, Japan).

### 2.4. Blood Parameters Including PUFAs

The serum EPA, DHA, AA, and DGLA levels were measured by an external laboratory (LSI Medience Corporation, Tokyo, Japan). The serum specimens were separated, frozen and stored at −80 °C until used.

A gas chromatography method (LSI Medience Corporation, Tokyo, Japan) was used for the assay of serum fatty acids as we reported previously [23].

### 2.5. Neutrophil Functions

BROS production and stimulated ROS (SROS) production were measured with flow cytometry (Becton Dickinson, San Jose, CA, USA) using the two-color method. ROS production was measured using the ROS-reacting fluorescent agent hydroethidine (HE; PolyScience Inc., Warrington, PA, USA). HE, a redox-sensitive probe, has been widely used to detect intracellular superoxide anion [23].

Hydroethidine (PolyScience Inc.) was adjusted to 44.4 μM and used as a ROS marker in neutrophils (HE reagent). As a stimulant, zymosan (OZ; Sigma-Aldrich, St. Louis, MO, USA), which is a yeast cell (Saccharomyces cerevisiae), opsonized with healthy adult pooled serum was adjusted to 5 mg/mL. As a hemolytic agent, a whole blood rising kit (Beckman Coulter Inc., Miami, FL) was used, and the hemolytic agent stock solution of the kit was diluted 25-fold with phosphate buffered saline (PBS) according to the instruction manual to prepare a required amount.

Two tubes for BROS and SROS were prepared for each person, and 100 μL of heparin-collected whole blood and 22 μL of HE reagent were mixed well and incubated at 37 °C for 5 min. After the incubation was completed, 25 μL of the stimulating reagent OZ was added to the SROS measurement tube, mixed well, and incubated at 37 °C for 35 min. After the incubation, 1.0 mL of the hemolytic agent was added and stirred for both SROS and BROS. Next, 250 μL of the fixative in the kit was added and stirred as soon as the solution became transparent, and the mixture was left standing for 5 min. After the hemolysis was completed, the cells were washed twice with PBS containing sodium azide, 50 μL of 5% paraformaldehyde was added at the end, and immediately measured by FACSCanto ll (Becton Dickinson and Company, Franklin Lakes, NJ, USA).

During flow cytometry, neutrophils were irradiated with a 488-nm laser beam generated from a 15-mW argon laser with forward- and side-scattering emission, which was simultaneously recorded. Green fluorescence generated from fluorescein isothiocyanate was detected through a 530-nm filter, and orange fluorescence generated from HE was detected through a 585-nm filter. Fluorescence intensity (FI) was measured as the value of neutrophils per 10,000 screened from the forward- and side-scattering emission for each sample. Cumulative FI, i.e., the sum of the values of FI multiplied by the percentage of positive cells, was used as a quantitative index.

In this study, the amount of superoxide production was used as an index of ROS production. Superoxide is the upstream substance of ROS metabolism, and all of the ROS are metabolites of superoxide. Accordingly, the amount of superoxide production is considered to be the reflection of the entire production of ROS.

### 2.6. Statistics

All variables were graphically inspected for normality using histograms. Continuous variables are given as mean ± standard deviation (SD). Tukey’s honestly significant difference test was conducted on patient characteristics, concentration of each PUFA, and neutrophil function (BROS and SROS). The Pearson correlation was used to determine the relationship between neutrophil functions and PUFAs. A multiple regression model with the 2 different neutrophil functions, 5 PUFAs, age by group, and sex, was used for predictive analysis. The selection of independent variables was based on previous literature and included age, BMI, estradiol, and smoking habit [24,25,26,27,28,29]. Therefore, we controlled for these confounders to clarify the relationship between PUFA and neutrophil function. The threshold for significance was *p* < 0.05. All statistical analyses were conducted using SPSS version 24.0 (IBM Corporation, Armonk, NY, USA).

## 3. Results

### 3.1. Characteristics of the Study Subjects

Figure 1 shows the flowchart of the study participants. In total, 351 men (mean age 50.4 ± 15.2 y) and 544 women (mean age 54.2 ± 14.7 y) were enrolled in the study (*N* = 895; range, 20–91 y). Table 1 summarizes the participant characteristics, PUFA levels, and neutrophil activity.

The mean values for AA and DGLA levels were significantly lower among the old-age male group (AA: 202.4 ± 47.1 μg/dL, *p* < 0.001; DGLA: 39.4 ± 13.3 μg/dL, *p* < 0.001) than those among the other male groups and higher among the middle-age female group (AA: 226.7 ± 47.7 μg/dL, *p* < 0.001; DGLA: 45.6 ± 13.8 μg/dL, *p* < 0.01) than those among other female groups. The mean values for EPA and DHA levels increased significantly with age for both sexes. The mean value for AA/DGLA ratio was lowest in the middle-age group and highest in the old-age group among males. The mean value for SROS levels was highest in the young-age group among all females.

### 3.2. Relationship between ROS and PUFAs (Correlation Coefficient)

Table 2 and Table 3 summarize the results of the single correlation between neutrophil function and PUFAs stratified by age group of the participants. The correlation analysis revealed significant positive correlations between AA and BROS levels in the young-age male and middle-age female groups. In the young-age female group, AA was positively correlated with BROS and SROS. In the same manner, DGLA levels correlated positively with the SROS level in all groups other than the old-age male group. Negative correlations were found between AA/DGLA ratio and SROS in the middle-age male group and young-age female group. EPA correlated positively with BROS in the young male group, middle-age female group, and old female group. In the old-age female group, a significant positive correlation was found between EPA and BROS.

### 3.3. Relationship between ROS and PUFAs (Multiple Regression Analysis)

Multiple regression analysis was performed, where the dependent variables were BROS and SROS, and the independent variables were age, BMI, estradiol level, and smoking habit. The results are shown in Table 4 and Table 5. Multiple linear analyses showed a significant positive correlation between AA and BROS levels in the young-age male group (β = 0.173, *p* = 0.047); AA and SROS in the middle-aged female group (β = 0.185, *p* = 0.042); DGLA and SROS in the young-age female group (β = 0.279, *p* = 0.002) and both middle-aged sexes (men: β = 0.225, *p* = 0.012 and women: β = 0.236, *p* = 0.019); EPA and BROS levels in the male young-age group (β = 0.187, *p* = 0.035) and the middle-aged female group (β = 0.210, *p* = 0.031). In the middle-aged male group, negative correlations were found between the AA/DGLA ratio and SROS (β = −0.222, *p* = 0.012). No significant correlation was found between PUFA and neutrophil function in the old-age groups for either sex.

## 4. Discussion

### 4.1. Relationship between ROS and PUFAs

This study aimed to evaluate the association between PUFAs, particularly DGLA levels and AA/DGLA ratio, and ROS production by neutrophil function according to age and sex in the general population. Our results show that significant correlations exist between PUFAs, particularly DGLA levels and AA/DGLA ratio and ROS production among young- and middle-age groups, while old-age groups showed no correlations between PUFA and neutrophil function. Significant negative correlations were found between AA/DGLA ratio and SROS among the middle-age males. This study adds to the literature regarding the association between neutrophil function and PUFAs in the general population, specifically analyzing the AA/DGLA ratio as a predictor of D5D activity according to age and sex. It has been reported that AA stimulates the generation of superoxide and induces cytotoxicity [30,31,32]. DGLA may produce two free radicals, 8-hydroxyoctanoic acid and heptanoic acid which are derived from peroxidation of cyclooxygenase-catalyzed DGLA [21,33]. Studies report that D5D activity can be estimated by AA/DGLA ratio and that D5D is negatively correlated with ROS [19,34].

This study supports previous findings that AA in serum decreases with age, while EPA and DHA levels in serum increase with age [35,36,37]. Most studies have not focused on correlations between DGLA level and AA/DGLA ratio in relation to age, but this study found that DGLA, as well as AA, were lower and AA/DGLA ratio was higher in the old age male group.

In the present study, we found correlations in multiple regression analyses between PUFAs and ROS production in the young and middle-age groups, though there was no correlation in the old-age group. This may be due to eating habits which reflect personal and social characteristics, which accumulate over time. Our results also suggest that serum PUFA levels have a greater impact on ROS production of younger people than those of older people, who are affected more by chronic inflammation and inflammatory disease, as evidenced by the fact that PUFAs were not correlated with ROS production in the old-age participants of either sex. BROS production, which is considered a reflection of oxidative stress under normal conditions, may represent internal ROS induced PUFA sensitivity. Unfortunately, the function of oxidative stress in vivo has not been fully elucidated. Although ROS signal biological damage, they are also known to have biological defense functions [38]. Therefore, the positive correlation between PUFA and ROS production indicates that PUFAs may increase oxidative stress and negatively affect the body. Simultaneously, the correlation can be interpreted as evidence of the body’s self-defense ability.

### 4.2. Study Limitations

Several limitations in this study must be noted. First, our study population was geographically limited to a district in Japan and, therefore, is not generalizable to all ethnicities. Second, limitations of the cross-sectional study are as follows; (1) that there is no evidence of the causal relationship between PUFA and neutrophil function, and (2) cross sectional study excludes people who develop the outcome but die before study. Despite these limitations, the strength of this study is that it adds knowledge regarding the relationship between PUFA, especially DGLA and AA/DGLA, and neutrophil function according age and sex in a general population.

## 5. Conclusions

PUFA levels are positively associated with ROS production, especially BROS, in young and middle-age people in the general population. In addition, AA/DGLA ratio may indicate SROS level, especially in middle-age men. If PUFAs are found to positively influence neutrophil function, controlled PUFA intake may be recommended from a young age. Further in vivo research on PUFA influence in humans is warranted.

## Figures and Tables

**Figure 1 nutrients-12-03222-f001:**
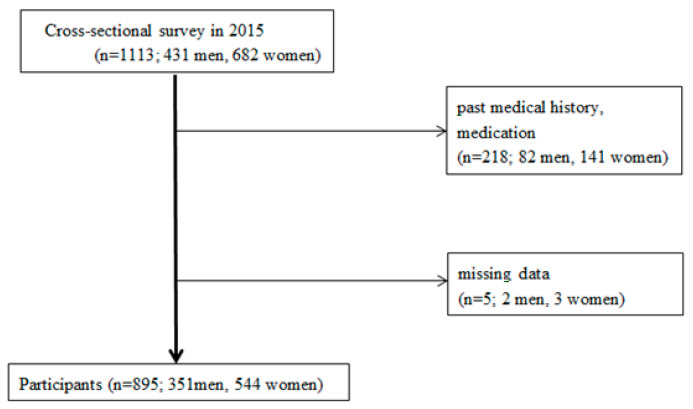
Flowchart of study participants.

**Table 1 nutrients-12-03222-t001:** Clinical characteristics of study participants.

	Male			Female		
	20–44 Years	44–64 Years	≥65 Years	20–44 Years	44–64 Years	≥65 Years
Age (year)	34.3 ± 5.7 *	54.3 ± 5.9 **	70.4 ± 5.6 ***	35.3 ± 5.9 †	55.8 ± 5.6 ††	70.9 ± 5.3 †††
Height (cm)	172.7 ± 5.4 *	169.2 ± 5.8 **	163.1 ± 8.4 ***	159.4 ± 5.4 †	156.1 ± 5.2 ††	151.0 ± 5.1 †††
Weight (kg)	70.1 ± 11.8	68.0 ± 8.8 **	62.0 ± 8.4 ***	54.5 ± 10.1	53.8 ± 7.8	51.9 ± 7.9 †††
BMI	23.4 ± 3.6	23.7 ± 2.7	23.3 ± 2.9	21.4 ± 4.0	22.0 ± 3.1	22.7 ± 3.1 †††
AA (μg/dL)	234.1 ± 56.4	236.2 ± 59.6 **	202.4 ± 47.1 ***	209.1 ± 40.5 †	226.7 ± 47.7 ††	206.4 ± 46.1
DGLA (μg/dL)	50.5 ± 15.7	53.4 ± 16.8 **	39.4 ± 13.3 ***	40.4 ± 14.6 †	45.6 ± 13.8 ††	41.3 ± 12.9
AA/DGLA	4.88 ± 1.27	4.66 ± 1.28 **	5.50 ± 1.53 ***	5.60 ± 1.62	5.29 ± 1.51	5.31 ± 1.46
EPA (μg/dL)	50.0 ± 26.5 *	82.1 ± 45.9 **	125.9 ± 62.5 ***	43.1 ± 22.8 †	82.6 ± 43.6 ††	117.6 ± 61.6 †††
DHA (μg/dL)	127.4 ± 45.5 *	163.5 ± 53.7 **	189.8 ± 64.3 ***	118.1 ± 36.2 †	161.3 ± 48.1 ††	194.0 ± 44.9 †††
BROS (×10^3^)	3.20 ± 3.40	4.10 ± 5.95	4.05 ± 6.95	3.28 ± 3.83	3.41 ± 6.69	3.34 ± 3.06
SROS (×10^6^)	7.52 ± 3.75	7.46 ± 3.24	7.67 ± 3.55	6.74 ± 3.28 †	5.81 ± 2.30	5.91 ± 2.91 †††

Tukey’s honestly significant difference test of between the three age cohorts. * †, compared 20–44 years with 44–64 years, *p* < 0.05; ** ††, compared 45–64 years with ≥65 years, *p* < 0.05; *** †††, compared 20–44 years with ≥65 years, *p* < 0.05. AA: arachidonic acid, BMI: body mass index, BROS: basal reactive oxygen species, DGLA: dihomo gamma linolenic acid, DHA: docosahexaenoic acid, EPA: eicosapentaenoic acid, SROS: stimulated reactive oxygen species.

**Table 2 nutrients-12-03222-t002:** Relationship between BROS and PUFAs (Correlation coefficient).

		AA	DGLA	AA/DGLA	EPA	DHA
**Male**						
young	r	0.171 *	0.135	−0.045	0.175 *	0.163
	*p* value	0.047	0.117	0.600	0.041	0.058
middle age	r	−0.140	−0.090	−0.012	−0.022	−0.073
	*p* value	0.109	0.300	0.891	0.802	0.404
old	r	−0.015	−0.083	0.101	0.091	0.045
	*p* value	0.892	0.458	0.365	0.416	0.686
**Female**						
young	r	0.220 **	0.081	0.041	−0.054	−0.030
	*p* value	0.005	0.308	0.606	0.496	0.703
middle age	r	0.162 *	0.054	0.065	0.142 *	0.097
	*p* value	0.014	0.417	0.328	0.032	0.146
old	r	0.041	−0.034	0.041	0.178 *	0.061
	*p* value	0.612	0.669	0.612	0.025	0.445

Pearson correlation coefficients between neutrophil BROS production (CFI) × Neutrophil counts, ×10^6^ and PUFAs (μg/mL). ** Correlation is significant at *p* < 0.01 * Correlation is significant at *p* < 0.05. AA: arachidonic acid, BROS: basal reactive oxygen species, CFI: cumulative fluorescence intensity, DGLA: dihomo gamma linolenic acid, DHA: docosahexaenoic acid, EPA: eicosapentaenoic acid, PUFAs: polyunsaturated fatty acids.

**Table 3 nutrients-12-03222-t003:** Relationship between SROS and PUFAs (Correlation coefficient).

		AA	DGLA	AA/DGLA	EPA	DHA
**Male**						
young-age	r	0.121	0.201 *	−0.154	0.022	0.081
	*p* value	0.161	0.019	0.074	0.799	0.347
middle-aged	r	0.090	0.266 **	−0.253 **	−0.134	0.019
	*p* value	0.300	0.002	0.003	0.124	0.829
old-age	r	−0.012	0.082	−0.169	0.001	−0.019
	*p* value	0.914	0.462	0.129	0.995	0.868
**Female**						
young-age	r	0.169 *	0.291 **	−0.231 **	−0.006	0.061
	*p* value	0.033	0.000	0.003	0.939	0.446
middle-aged	r	0.124	0.191 **	−0.123	0.038	0.142 *
	*p* value	0.062	0.004	0.063	0.566	0.032
old-age	r	0.140	0.185 *	−0.143	−0.044	0.069
	*p* value	0.080	0.020	0.073	0.585	0.389

Pearson correlation coefficients between neutrophil SROS production (CFI) × Neutrophil counts, ×10^6^ and PUFAs (μg/mL). ** Correlation is significant at *p* < 0.01 * Correlation is significant at *p* < 0.05. AA: arachidonic acid, CFI: cumulative fluorescence intensity, DGLA: dihomo gamma linolenic acid, DHA: docosahexaenoic acid, EPA: eicosapentaenoic acid, PUFAs: polyunsaturated fatty acids, SROS: stimulated reactive oxygen species.

**Table 4 nutrients-12-03222-t004:** Multiple linear regression analysis with BROS as dependent variable.

	Male						Female					
	Young-Age		Middle-Aged		Old-Age		Young-Age		Middle-Aged		Old-Age	
	β	*p*	β	*p*	β	*p*	β	*p*	β	*p*	β	*p*
AA	0.173	0.047 *	−0.126	0.159	0.003	0.978	0.128	0.128	0.122	0.200	0.259	0.076
DGLA	0.076	0.408	−0.051	0.566	−0.016	0.894	0.141	0.138	−0.085	0.423	−0.015	0.915
AA/DGLA	0.046	0.616	−0.054	0.539	0.006	0.962	−0.003	0.973	0.196	0.053	0.152	0.277
EPA	0.187	0.035 *	−0.040	0.654	0.052	0.644	0.031	0.710	0.210	0.031 *	0.190	0.192
DHA	0.102	0.258	−0.103	0.229	0.050	0.657	−0.075	0.940	0.108	0.281	0.260	0.068

Multiple regression analysis between neutrophil BROS production (CFI) × neutrophil counts and unsaturated fatty acids (μg/mL) β: standardized coefficient, * Statistical significant at *p* < 0.05, Forced entry multiple regression was performed by gender using the BROS production as the dependent variable and age, BMI, estradiol and habit of smoking as independent variables. AA: arachidonic acid, BROS: basal reactive oxygen species, CFI: cumulative fluorescence intensity, DGLA: dihomo gamma linolenic acid, DHA: docosahexaenoic acid, EPA: eicosapentaenoic acid, PUFAs: polyunsaturated fatty acids.

**Table 5 nutrients-12-03222-t005:** Multiple linear regression analysis with SROS as dependent variable.

	Male						Female					
	Young-Age		Middle-Aged		Old-Age		Young-Age		Middle-Aged		Old-Age	
	β	*p*	β	*p*	β	*p*	β	*p*	β	*p*	β	*p*
AA	0.092	0.274	0.059	0.517	0.003	0.979	0.145	0.068	0.185	0.042 *	0.215	0.129
DGLA	0.098	0.269	0.225	0.012 *	0.095	0.427	0.279	0.002 *	0.236	0.019 *	0.156	0.258
AA/DGLA	−0.056	0.521	−0.222	0.012 *	−0.158	0.164	−0.059	0.483	−0.100	0.305	−0.082	0.545
EPA	−0.010	0.912	−0.100	0.270	0.017	0.880	0.019	0.806	0.062	0.507	−0.020	0.889
DHA	0.041	0.635	0.066	0.451	−0.061	0.587	0.153	0.053	0.145	0.130	0.222	0.105

Multiple regression analysis between neutrophil SROS production (CFI) × neutrophil counts and unsaturated fatty acids (μg/mL) β: standardized coefficient, * Statistical significant at *p* < 0.05, Forced entry multiple regression was performed by gender using the BROS production as the dependent variable and age, BMI, estradiol and habit of smoking as independent variables. AA: arachidonic acid, CFI: cumulative fluorescence intensity, DGLA: dihomo gamma linolenic acid, DHA: docosahexaenoic acid, EPA: eicosapentaenoic acid, PUFAs: polyunsaturated fatty acids, SROS: stimulated reactive oxygen species.

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
