# Peer review of "Association between Polyunsaturated Fatty Acid and Reactive Oxygen Species Production of Neutrophils in the General Population"

_nutrients, 2020, doi:10.3390/nu12113222_

Round 1

Reviewer 1 Report     

This report shows the correlation analysis between PUFAs levels and ROS produced by neutrophils in different-age and gender populations, and found  DGLA and AA/DGLA ratios are positively associated with neutrophil ROS level  in the young and middle-aged groups. This manuscript should be further strengthened by addressing a few concerns as follow:

When testing the ROS level of neutrophil in whole blood, do authors gate out the neutrophils from other leukocyte subsets? Do other leukocyte subsets (monocytes?) also produce ROS? Can the authors show the gating strategy for neutrophil? do the authors gate out dead cells?

Author Response

Response to Reviewer 1 Comments

Point 1: This manuscript should be further strengthened by addressing a few concerns as follow:

When testing the ROS level of neutrophil in whole blood, do authors gate out the neutrophils from other leukocyte subsets? Do other leukocyte subsets (monocytes?) also produce ROS? Can the authors show the gating strategy for neutrophil? do the authors gate out dead cells?

Response 1: Thank you for providing these insights. We reported on ROS production of neutrophils in previous studies. The flow cytometry can detect neutrophils in white blood cells based on the size of cells and the characteristics of their internal structure. In this study, only neutrophils are measured. The neutrophil-producing ROS was evaluated as follows. Cumulative FI, i.e., the sum of the values of FI multiplied by the percentage of positive cells, was used as a quantitative index. We have not measured the ROS production of dead cells or monocytes basically. I'm not a native speaker in English, so I'm a little worried if the reviewers' comments are answered correctly. If so, please point it out and give a chance to re-review. Thank you once again for your valuable comments and suggestions.

Reviewer 2 Report

The study by Suzukietal, describes the association of PUFA (analyzed by gas chromatography, in particular, DGLA and AA/DGLA) and  ROS (with flow cytometry)  production by neutrophils in a population of 895 participants to the study extracted from a district in Japan.

The neutrophil functions considered were basal ROS production assumed a reflection of oxidative stress under normal condition, and stimulated ROS production, intended as a reflection of oxidative stress under stimulation

The participants were characterized clinically and divided into groups (male/female and range of Years)

The authors found a positive association between PUFA levels and ROS production in particular in young and middle-age people

Although, as the authors themselves state, there is no evidence of a causal relationship between PUFA and neutrophil function, the study is of interest as it can consider a basis for a starting point to take into account in case of PUFA supplementation.

Specific comment

Line 127-128: please clarify the sentence

Line 91-95 is a  repeat of line 85-89

References are adequate

Author Response

Response to Reviewer 2 Comments

 Point 1: Line 127-128: please clarify the sentence

 Response 1: Thank you for your suggestion. I deleted last sentence on line 128.

 Point 2: Line 91-95 is a repeat of line 85-89.

 Response 2: I deleted line 91-95. I'm not a native speaker in English, so I'm a little worried if the reviewers' comments are answered correctly. If so, please point it out and give a chance to re-review. Thank you for taking the time and energy to help us improve the paper.